# Role of Transglutaminase 2 in Cell Death, Survival, and Fibrosis

**DOI:** 10.3390/cells10071842

**Published:** 2021-07-20

**Authors:** Hideki Tatsukawa, Kiyotaka Hitomi

**Affiliations:** Cellular Biochemistry Laboratory, Graduate School of Pharmaceutical Sciences, Nagoya University, Tokai National Higher Education and Research System, Nagoya 464-8601, Aichi, Japan; hitomi@ps.nagoya-u.ac.jp

**Keywords:** transglutaminase, crosslinking, TG2, cell death, cell survival, macrophage activation, fibrosis

## Abstract

Transglutaminase 2 (TG2) is a ubiquitously expressed enzyme catalyzing the crosslinking between Gln and Lys residues and involved in various pathophysiological events. Besides this crosslinking activity, TG2 functions as a deamidase, GTPase, isopeptidase, adapter/scaffold, protein disulfide isomerase, and kinase. It also plays a role in the regulation of hypusination and serotonylation. Through these activities, TG2 is involved in cell growth, differentiation, cell death, inflammation, tissue repair, and fibrosis. Depending on the cell type and stimulus, TG2 changes its subcellular localization and biological activity, leading to cell death or survival. In normal unstressed cells, intracellular TG2 exhibits a GTP-bound closed conformation, exerting prosurvival functions. However, upon cell stimulation with Ca^2+^ or other factors, TG2 adopts a Ca^2+^-bound open conformation, demonstrating a transamidase activity involved in cell death or survival. These functional discrepancies of TG2 open form might be caused by its multifunctional nature, the existence of splicing variants, the cell type and stimulus, and the genetic backgrounds and variations of the mouse models used. TG2 is also involved in the phagocytosis of dead cells by macrophages and in fibrosis during tissue repair. Here, we summarize and discuss the multifunctional and controversial roles of TG2, focusing on cell death/survival and fibrosis.

## 1. Introduction

Transglutaminases (TGase) are multifunctional enzymes and constitute a family of eight isozymes designated as blood coagulation factor XIII and TG1–7. In this family, TG2 is widely distributed and involved in multiple biological processes. It catalyzes a Ca^2+^-dependent acyl transfer reaction between the γ-carboxamide group of glutamine present in a particular sequence and either primary amines, such as polyamines and histamine, or the ε-amino group of a lysine residue, intra- or inter-molecularly. Water replaces the amine donor substrates, leading to the deamidation of glutamine. In addition, TG2 and factor XIIIa exhibit a Ca^2+^-dependent isopeptidase activity and can hydrolyze the isopeptide bond, at least under test tube conditions. TG2 exerts additional enzymatic activities that do not require Ca^2+^, i.e., it hydrolyzes ATP and GTP to mediate signal transduction through G-protein-coupled receptors, protein disulfide isomerases, and protein kinases, as well as interacts with several proteins as an adhesion or scaffold protein [1] (Figure 1).

TG2 is ubiquitously distributed inside (in the nucleus, cytoplasm, plasma membrane, and mitochondria) and outside the cell, where it appears in the extracellular matrix (ECM) and exosome [2,3,4]. In mammals, TG2 is detected across the body, including the blood, extracellular spaces, and intracellular compartments of nearly all tissues. It is involved in cell death, growth, and differentiation as well as tissue repair by tissue remodeling/wound healing and ECM assembly [5]. In this article, we focus on the role of TG2 in cell death, macrophage activation, and tissue repair processes, which are involved in several pathogeneses, including tissue injury, inflammation, and fibrosis. This review aims to summarize the recent knowledge on the mechanisms activated by TG2 to regulate cell death/survival and fibrosis in the tissue repair process.

## 2. Multifunctional Activity and Regulation of TG2

### 2.1. Crosslinking (Transamidase) Activity

When cells are exposed to increased intracellular Ca^2+^ concentrations (>700–800 nM) in response to certain stimuli, including injury and inflammation signals, TG2 structural conformation is dramatically altered and changes from a closed to an opened form that exerts crosslinking and transamidase activities [6,7]. In the appropriate redox condition, a TG2 intermediate thioester is formed through the attack of an acyl donor (γ-carboxamide group of a protein glutamine residue) by the nucleophilic active thiolate (cysteine residue at the active site of TG2), with release of ammonia. Then, the thiolate is restored via the nucleophilic attack of an acyl acceptor (ε-amino group of a protein lysine residue), leading to the formation of a covalent intra- or inter-molecular *N*^ε^-(γ-glutamyl)lysine isopeptide bond, which is resistant to degradation [8]. It has been suggested that the isopeptide bond contributes to the stabilization of the ECM and the prevention of the release of the intracellular content from apoptotic cells into the extracellular milieu. A similar reaction also occurs through the incorporation of primary amines and polyamines into the γ-carboxamide group of a protein glutamine residue.

### 2.2. Deamidation Activity

If the aforementioned intermediate thioester bond is attacked by a water molecule as the acyl acceptor, a deamidation reaction occurs, in which the site-specific acyl-donor glutamine is converted to a glutamate residue [3]. For many years, the deamination reaction was believed to occur as a side reaction of the absence of primary amines or at low pH, when amine availability was limited. In these conditions, water would play a role owing to its abundance [9,10]. However, site-specific deamidations of heat shock protein [11] and βB2/3-crystallines [12] have been reported, suggesting that the substrate affinity for TG2 and the reaction conditions influence the propensity toward deamidation or transamidation [13].

### 2.3. GTPase and ATPase Activities

Aside from its transamidase activity, TG2 possesses several other enzymatic functions independent of Ca^2+^. When the intracellular Ca^2+^ concentration is as low as 10–20 nM, TG2 binds and hydrolyzes GTP and ATP, participating in the transmembrane signaling of phospholipase Cδ as a component of α1B/α1D adrenergic, thromboxane A2, and oxytocin receptors [4]. The transamidase and GTPase activities are mutually exclusive, whereas ATP binding has no effect on the transamidase activity. The GTP-binding form of TG2 sustains the closed conformation, which prevents the formation of the Ca^2+^-binding open form exerting the transamidase activity and vice versa.

### 2.4. Isopeptidase Activity

The reversible crosslinking of α2-plasmin inhibitor to fibrinogen and fibrin by factor XIIIa was reported and is potentially involved in the regulation of fibrinolytic processes [14,15]. Biochemical studies demonstrated that TG2 also exhibits an isopeptidase activity targeting *N*^ε^-(γ-glutamyl)lysine [16]. Therefore, an unknown regulatory system of TG2 might exist to separately switch on or off the transamidase and isopeptidase activities. Specific TG2 mutants, which exhibit deficient transamidase (W332F) and isopeptidase (W278F) activities, have been identified [17]. Further research might help elucidate the role of the TG2 isopeptidase activity in physiological and pathological processes.

### 2.5. Adapter/Scaffold Activity (Enhanced Integrin–Fibronectin Interaction)

TG2 has originally been investigated as a cytosolic protein. However, it is secreted on the cell surface and in the extracellular space. It has been reported that TG2 promotes the stabilization and deposition of ECM proteins through its crosslinking activity [18,19,20,21]. In addition, TG2 forms a heterocomplex with fibronectin and interacts with integrins and heparan sulfate proteoglycans in a crosslinking activity-independent manner [22,23]. The TG2–fibronectin complex promotes fibril formation and RGD (arginine–glycine–aspartic acid)-independent focal adhesion through syndecans and integrins [23]. Moreover, it contributes to cell survival in osteoblasts [24], mesenchymal stem cells [25], and various tumor cells [26,27].

### 2.6. Other Functions

TG2 was reported to demonstrate a protein disulfide isomerase (PDI) activity, which has been implicated in mitochondrial-dependent apoptosis [28,29]. TG2 also exerts an intrinsic serine/threonine kinase activity to phosphorylate insulin-like growth factor binding protein-3 [30], p53 tumor suppressor protein [31], histones H1–4 [32], and retinoblastoma (Rb) protein [33]. Furthermore, TG2 affects hypusine metabolism, regulating the activity of eukaryotic initiation factor 5A and cell proliferation [34]. Recently, TG2 was reported to serotonylate histone H3 trimethylated lysine 4 (H3K4me3)-marked nucleosomes, controlling the recruitment of transcription factors, including TFIID [35].

## 3. Regulation of TG2 Expression and Activity

### 3.1. Regulation of TG2 Expression

*TGM2* gene expression is regulated by various cellular events, including apoptotic stimuli [36,37], viral infection [38], endoplasmic reticulum (ER) stress [39,40], hypoxia/ischemia [41,42,43], inflammation [44], tissue remodeling [45], and cancer [46,47,48]. It is mediated by several related factors and cytokines, such as retinoids [49,50], lipopolysaccharides [51], transforming growth factor (TGF)-β/bone morphogenetic protein 4 [52], nuclear factor-κ B (NF-κB) [48,53], glucocorticoids [54], interleukin (IL)-1 [55], IL-6 [56], hypoxia-inducible factor-1 [57], tumor necrosis factor (TNF)-α [58], and epidermal growth factor (EGF) [46]. Retinoic acid is a well-known inducer of TG2 expression and promotes the cellular differentiation of neutrophil granulocytes [59,60] and neuroblastoma cells [61,62] through the heterodimer retinoid acid receptor (RAR)/retinoid X receptor (RXR) and transcription factor Sp1 [50]. TG2 expression is upregulated in cancer cells resistant to chemotherapy or with high metastatic potential. *TGM2* promoter contains response elements to inflammation and hypoxia, which are greatly elevated in the environment surrounding malignant tumors, leading to an increased expression of TG2 [57]. Ischemia also promotes TG2 expression [41]. In addition, n-Myc and c-Myc contribute to the regulation of TG2 expression by recruiting histone deacetylase 1 protein to the *TGM2* promoter in cancer cells [63]. Interestingly, the antiproliferative effects of histone deacetylase inhibitors in cancer cells are impaired by the induction of TG2 mRNA and protein expression, suggesting that TG2 is involved in the resistance of cancer cells [64]. Since the half-life of TG2 is about 10 h in colorectal cancer HT29 cells, the sustained protein synthesis of TG2 is necessary for cancer proliferation and resistance to anticancer drugs such as histone deacetylase inhibitors [64]. TG2 influences TGF-β activation and signaling, whereas TGF-β1 was reported to promote and suppress TG2 expression [52,65]. TG2 expression is increased by several cytokines, such as IL-6, TNF-α, and NF-κB, in many cell types, including human hepatoblastoma cells [56,58] and macrophages [66,67]. AF4/FMR2 family member 1, known as a central scaffolding protein of super elongation complex, was recently reported to contribute to TG2 expression after being recruited to the *TGM2* promoter in mouse adipocytes [68].

Human and murine *TGM2* promoters are well characterized and contain various response elements for retinoic acid (−1731 bp and −1720 bp), glucocorticoids (−1399 bp), NF-κB (−1338 bp), IL-6 (−1190 bp), TGF-β1 (−900 bp), estrogens (−656 bp) [69,70], activator protein-2 (AP-2, −634 bp), AP-1 (−183 bp), hypoxia (−367 bp), and nuclear factor-1 (+4 bp, +12 bp) [50,71], as well as motif regions, such as CAAT box (−96 bp), TATA box (−29 bp), and GC box, for Sp1 binding (−54 bp, −43 bp, +59 bp, +65 bp) [72]. Figure 2A presents the regulatory elements previously reported for the human *TGM2* promoter. In addition, TG2 expression is directly downregulated by micro-RNA 19, which is responsible for the increased invasion and metastasis of colorectal cancer cells [73]. In addition, enhancer RNA molecules of TG2 expression were identified to regulate the recruitment of the transcriptional repressor CTCF in the intergenic region of thymocytes treated with retinoids and TGF-β to induce their death [74]. Furthermore, a recent review summarized the various potential binding sites of transcription factors and single-nucleotide polymorphisms of the *TGM2* promoter using information obtained from the public database of chromatin immunoprecipitation sequencing [75]. Finally, several splicing variants of *TGM2* exhibit different regulatory properties and catalytic activities, affecting the global TG2 activity [76]. Therefore, *TGM2* gene expression is regulated by multiple signaling pathways involved in physiological and pathological events, although these functional roles have not been fully elucidated.

### 3.2. Regulation of TG2 Activity

A few studies have focused on the regulation of TG2 activity compared with the number of investigations on its transcriptional regulation. Ca^2+^ and GTP are known as a competitive activator and a suppressor of TG2 transamidase activity, respectively. Ca^2+^ binding alters TG2 structural conformation by moving the β-barrel domains 3 and 4 away from the catalytic domain 2, opening the active center and facilitating access to the substrate. TG2 appears to be inactive in cells in the absence of stress (~100 nM free cytoplasmic [Ca^2+^]). In addition, a free cytoplasmic GTP concentration higher than 100 μM is required to maintain the closed GTP-bound conformation of TG2, which inhibits its transamidase activity [77].

The conditions in the extracellular space are suitable for TG2 activation as Ca^2+^ and GTP are present at high and low levels, respectively. However, a highly oxidative state was reported to keep TG2 in the inactive state in the absence of stress due to the formation of disulfide bonds as a posttranslational modification [78], which was reversed by thioredoxin-mediated reduction [79]. Furthermore, extracellular TG2 can be negatively regulated by S-nitrosylation, indicating that nitric oxide is also a potent inhibitor of TG2 activation and might be involved in age-related vascular stiffness [80,81]. TG2 acetylation was also reported to suppress its activity in vitro [82]. Finally, TG2 is stabilized by SUMOylation, which inhibits TG2 ubiquitination, leading to enhanced protein levels and activity [83,84].

The interaction between TG2 and membrane lipids might be another regulatory factor of TG2 transamidase activity. Sphingosylphosphocholine reduces the Ca^2+^ requirement for TG2 activation, which might allow TG2 transamidase activity resulting from the conformational changes induced by locally increased Ca^2+^ levels [85]. The alternative splicing of *TGM2* is also involved in the regulation of the transamidase activity. Indeed, C-terminal-truncated variants lack part of or the entire GTP-binding pocket. Therefore, it is expected that the transamidase activity is not suppressed even by high GTP concentrations, resulting in an increased sensitivity and level of TG2 activation by Ca^2+^ under physiological conditions. Figure 2B presents the previously reported regulatory factors of TG2 activity.

## 4. TG2 Functions in Cell Death and Survival

The multifunctional activity, genetic variants, and conformational changes of TG2 complicate the understanding of its role in physiological and pathological events. The various functions of TG2, i.e., as a GTPase, PDI, kinase, and adapter/scaffold, and its role in transamidase activity are associated with both cell death and survival in various cellular environments. In addition, TG2 is localized in several subcellular spaces, such as the ECM, plasma membrane, cytosol, mitochondria, recycling endosomes, and nucleus, which influence its biological activities (Figure 3).

TG2 accumulation has been demonstrated in various cellular and tissue types undergoing cell death [86]. For example, initial studies investigating the relationship between TG2 and cell death indicated that an enhanced TG2 crosslinking activity was correlated with the extent of cell death [87], whereas TG2 inhibition reduced apoptosis [88]. Transfection of TG2, but not of mutated TG2 lacking crosslinking activity, enhances caspase-dependent cell death [89]. TG2-induced cell death was also associated with the release of both cytochrome c [36] and apoptosis-inducing factor [90] from the mitochondria.

Previously, we reported that crosslinking of transcription factors by nuclear TG2 caused caspase-independent cell death. TG2 crosslinks and inactivates the general transcription factor Sp1, which results in a reduced expression of growth factor receptors, such as c-Met and EGF receptors, which are essential for cell survival [91,92]. Another group also demonstrated that TG2 polymerizes and inactivates Rb protein, which inhibits its interaction with E2F1 and enhances its degradation, accelerating cell growth arrest/apoptosis [93]. Contrarily, in fibroblast cells treated with retinoids, TG2 prevents Rb protein degradation by caspase-7 probably through its GTP-binding activity, leading to an attenuation of apoptosis [94].

By investigating liver diseases, we previously demonstrated that TG2 transamidase activity significantly increases in the nucleus of hepatocytes treated with alcohol or free fatty acid, promoting the crosslinking and inactivation of Sp1. The defect in Sp1 activity causes the downregulation of the hepatocyte growth factor c-Met, leading to caspase-independent hepatic cell death in cultured hepatocytes and animal models as well as in patients with alcoholic and non-alcoholic steatohepatitis [39,91]. This proapoptotic role of TG2 crosslinking activity was investigated by other groups in carbon tetrachloride- (CCl4) and ethanol-induced liver injury, non-alcoholic fatty liver disease, and acute human liver injury [53,95,96,97,98,99]. However, a prosurvival role of TG2 has also been reported and attributed to both GTP-binding and crosslinking activities [100]. TG2 provides protection against liver injury, as the injuries induced by CCl4 or anti-Fas antibody are more severe in TG2-deficient mice than in wild-type controls [101,102]. These reports are inconsistent with our previous work. We speculate such an inconsistency to be caused by differences in the stimulant reagent doses and mice genetic backgrounds [91,103].

The relationship between TG2 and cell death has been investigated in several neuronal models [104]. TG2 expression is increased in the human brain in various chronic or acute neuropathological conditions [105]. Enhanced TG2 activity and expression are observed in the ischemic hippocampus after reperfusion [106,107] and cultured astrocytes exposed to oxidative stress [108], leading to neurodegeneration. TG2-deficient mice or those treated with a TG2 inhibitor present a smaller infarction volume after reperfusion than control mice [109]. However, controversial evidence indicated that TG2 might play a protective role in response to stress. In hypoxic conditions induced by ischemia and stroke [41,110,111], TG2 binds to hypoxia-inducible factor 1β independently of its transamidase activity and prevents the upregulation of proapoptotic factors, such as Bnip3 [112] and Noxa [113], thereby preventing neuronal cell death.

Transgenic mice overexpressing human TG2 selectively in neurons exhibited a dramatic increase in neuronal damage in the sensitive hippocampal regions after treatment with kainic acid, even though these mice presented no apparent phenotype in the absence of stress [114]. Enhanced TG2 expression and/or activity, especially of nuclear TG2, has been observed in several neurodegenerative disorders, such as Alzheimer’s disease (AD), Huntington’s disease (HD), and Parkinson’s disease (PD) [115]. TG2 was reported to crosslink both amyloid-β peptide and tau protein in vitro [116,117]. The resultant polyaminated tau protein is more resistant to proteolytic degradation by calpain, which indicates that TG2 may contribute the aggregation processes of amyloid-β and tau in AD patients [118]. In addition, the levels of truncated alternative spliced TG2 variants lacking GTPase activity are also enhanced in AD patients and possess potential proapoptotic properties [119,120]. In the frontal cortex of postmortem HD brains, 99% colocalization is observed between *N*^ε^-(γ-glutamyl)lysine crosslinks and huntingtin aggregates in the nucleus [121], indicating the involvement of nuclear TG2 in HD. Furthermore, TG2-deficient HD mouse models experience a significant delay of motor dysfunction onset and a prolonged survival. TG2 inhibitors also ameliorated HD symptoms via transcriptional dysregulation [122,123]. Mutant huntingtin binds to other polyglutamine-enriched proteins, such as transcription factors, including Sp1 or its coactivator TAFII130 [124,125,126], and interferes with their inactivation. This might repress the Sp1-mediated expression of prosurvival factors and metabolic-related genes, such as brain-derived neurotrophic factor [127], dopamine D2 receptor [124,125], preproenkephalin [124], peroxisome proliferator-activated receptor-γ coactivator-1α [122], and cytochrome c [122]. These results indicate that TG2 is potentially an important factor aggravating HD symptoms through the transcriptional dysregulation of several survival factors and key metabolic genes, although TG2 is not critical for inducing HD. Moreover, normal huntingtin protein localizes to nuclear actin–cofilin rods during stress and is required for a proper stress response involving actin remodeling. Defective nuclear actin remodeling leads to faster cell death and is correlated with disease progression [128]. It was associated with mutant huntingtin, and stress-activated TG2 crosslinks actin–cofilin in HD, leading to neurodegeneration. TG2 appeared to be highly expressed in the substantia nigra of PD patients and colocalized with α-synuclein, a potential substrate for TG2 in vivo [129], in the brain of patients with dementia with Lewy bodies [130], and mediated toxicity of α-synuclein in vivo [131].

In addition to the regulation of cell death and survival, TG2 is involved in the activation of several immune cells such as dendritic cells [132], T cells [133,134,135], B cells [136], macrophages [137], and neutrophils [60]. The intake of deamidated gluten modified by TG2 through food also causes an adaptive immune response in celiac disease patients, accompanied by massive cell death in small intestinal epithelial cells [138,139,140]. The relationship between celiac intestinal barrier defect and hepatitis has been reported [141], and it is thought that an amplifying loop in liver diseases is initiated, with cytokine secretion by hepatocytes and consecutive intestinal barrier defect [142]. With regard to the clearance of apoptotic cells, TG2 promotes dimerization of the monocyte chemotactic factor S19 and consequently monocyte infiltration [143]. Furthermore, a defective clearance of apoptotic cells or lipids by macrophages in the thymus and Kupffer cells in the liver of TG2-deficient mice has been reported [144,145,146]. Because these mice still demonstrated inflammatory infiltration of macrophages at the apoptosis sites and developed autoimmunity, TG2 was proposed to be required for the engulfment of apoptotic cells by macrophages but not for their recognition and binding. Subsequent studies demonstrated that the role of TG2 in phagocytosis depends on GTP-binding sites but not on its transamidase activity [147,148]. In addition, TG2 contributes to the formation of a complex with milk fat globule-EGF factor 8 and integrin β3 on the surface of macrophages and microglia and thus was required for the formation of engulfing portals [148,149]. These relationships between TG2 and macrophages have been well summarized in Kaartinen’s review [150].

## 5. TG2 Functions in Fibrosis

The TG2 transamidase activity contributes to the wound healing process and fibrosis. The reaction products form an *N*^ε^-(γ-glutamyl)lysine isopeptide bond resulting from the crosslinking. This is an important step for the maturation and stabilization of ECM components, such as collagens, exacerbating scarring and fibrosis in various tissues, including the liver [96,151,152,153], kidney [154,155,156,157,158,159], lung [160,161,162,163,164,165], and heart [166,167]. The other enzymatic crosslinker, lysyl oxidase (LOX), has also been reported to contribute to collagen stabilization. LOX oxidizes certain lysine residues in collagen to produce aldehydes, which react to form covalent bonds and stabilize molecules within the collagen fibers [168]. Impaired crosslinking by LOX results in weak collagen fibers and fragile collagenous tissue [169]. In the remodeling of fibroblast-populated collagen lattices, TG2 predominantly contributes to the Ca^2+^-dependent early entrenchment (initial remodeling) by crosslinking of the extant matrix, whereas LOX implicates Ca^2+^-independent contractility at later times [170]. These results suggest that, in fibrosis, TG2 is involved in early ECM remodeling, while LOX contributes to subsequent modification.

Aside from the direct ECM stabilization, the TG2 transamidase activity appears to play a significant role in the fixation and activation of the profibrotic cytokine TGF-β. TGF-β is released in a latent form and converted to an active one. Enhanced TG2 activity is required for TGF-β activation from the latency binding complex as it promotes crosslinking of the large latent TGF-β binding protein to fibronectin or other ECM components on the cell surface [165,171,172,173,174]. The secretion of TG2 into the ECM is important for its function in TGF-β activation. The secretion mechanisms are unclear, as TG2 lacks the signal peptide necessary for ER targeting and the classical protein secretion mechanism through the ER–Golgi system. Moreover, no Golgi-associated protein modification, such as glycosylation, has been evidenced for TG2 [175]. Recent studies demonstrated that TG2 interacts with the heparan sulfate chains of proteoglycans, forms a complex with fibronectin, and interacts with integrins and heparan sulfate proteoglycans in the ECM to promote cell adhesion and spreading [176,177]. The interaction between TG2 and the heparan sulfate chains of cell surface syndecans is a potential mechanism implicated in the pathophysiological role of TG2, including in fibrosis [23,178,179].

As previously described, we demonstrated that nuclear TG2 inactivated Sp1 by crosslinking, leading to reduced expression of c-Met and consequently activation of hepatic apoptosis in a hepatic injury mouse model and in patients with alcoholic steatohepatitis [91]. TG2-mediated reduction of c-Met expression might be involved in the impaired hepatocyte regeneration observed in patients with alcoholic liver diseases [103,180,181]. Furthermore, hepatocyte-specific c-Met-deficient mice demonstrated more extensive liver cell damages and fibrosis, indicating that the induction of nuclear TG2/crosslinked Sp1/downregulated c-Met axis accompanied liver fibrosis. In agreement with our findings, TG2 nuclear accumulation and crosslinked Sp1 were observed in the fibrotic area of patients with alcoholic steatohepatitis [182].

However, TG2-deficient mice demonstrated no alteration of the fibrosis levels in the liver after treatment with CCl4 or thioacetamide [183]. This contradictory result might be due to discrepancies in the method used to target the *TGM2* gene, in the mouse background, and in the disease model. We obtained similar results using TG2-deficient mice. Indeed, liver fibrosis induced by bile duct ligation was not inhibited in these mice [152], although these mice presented a significant reduction of fibrosis induction in other fibrosis models, such as kidney fibrosis induced by unilateral ureteral obstruction [158] and lung fibrosis resulting from bleomycin treatment [162]. Interestingly, TG2 or pan-TGase inhibitors, including competitive or reversible/irreversible inhibitors, have been demonstrated to be consistently protective in several fibrosis models, including liver fibrosis induced by both CCl4 and bile duct ligation [98,152,158,184,185,186,187,188,189].

## 6. Conclusions and Prospects

Since the TGase family is multifunctional and contains numerous isozymes and splicing variants, an integrated understanding of TGase functions in pathophysiological events is often difficult to achieve. The role of TG2 in cell death and survival remains controversial. However, TG2 may be generally involved in the positive effect of apoptotic cell phagocytosis by macrophages and fibrosis induction. At low Ca^2+^ levels and high GTP concentrations in normal cellular condition without stress, TG2 exists as a closed form and exerts GTPase, PDI, and kinase activities to maintain homeostasis, which is crucial for cell survival. Contrarily, abnormal cellular conditions with stress and high Ca^2+^ concentrations allow TG2 change of conformation to the open form, which exhibits crosslinking activity, leading to alterations of its subcellular localization and extracellular release. Activated TG2 in the cytoplasm or nucleus of stressed cells crosslinks and regulates the activity and proteostasis by ubiquitination and autophagy of a number of target substrate proteins involved in several signaling events. Nuclear TG2 mainly regulates the activity of transcription factors and chromatin remodeling, which are involved in the expression of various downstream proteins important in cell death and survival.

In addition to the activation of TG2 during cell death induction, TG2 is upregulated in macrophages and is important for the clearance by phagocytosis of dead cells. After or following cell death and inflammation events, TG2 is secreted into the extracellular space and contributes to the enhancement of fibrogenesis, which allows filling the gaps resulting from cell death. In the ECM, TG2 forms a complex with fibronectin and interacts with integrins and heparan sulfate proteoglycans, contributing to the stabilization of the ECM mediated by crosslinking and to sustained TGF-β activation, leading to the development of fibrosis linked to organ dysfunctions (Figure 3).

Treatment with inhibitors of the TG2 crosslinking activity appeared to consistently suppress pathogenic fibrosis, although their effects on cell death and survival are not consistent, probably due to different experimental conditions and stimuli. Considering the ubiquitous and multifunctional nature of TG2, the development of a drug with no side effects might be difficult. Therefore, clinical compounds and antibodies that are more specific and allow controlling the drug distribution in the whole body are currently being developed. Effective TG2 inhibitors were developed by Zedira GmbH and are now in advanced clinical trials for the treatment of celiac disease. Recently, novel candidates for the treatment of kidney fibrosis were also successfully developed. In the future, these TG2 inhibitors will also be tested in patients with fibrosis in other organs such as lung and liver.

In our recent work, we conducted a comprehensive analysis of TG2-mediated crosslinking of substrate proteins in models of fibrosis targeting the liver, kidney, and lungs. Based on these results, we will create a database of the substrate proteins crosslinked in fibrosis, specifically, in each organ, which will support the development of novel preventive drugs against fibrosis acting by suppressing the crosslinking of substrate proteins.

## Figures and Tables

**Figure 1 cells-10-01842-f001:**
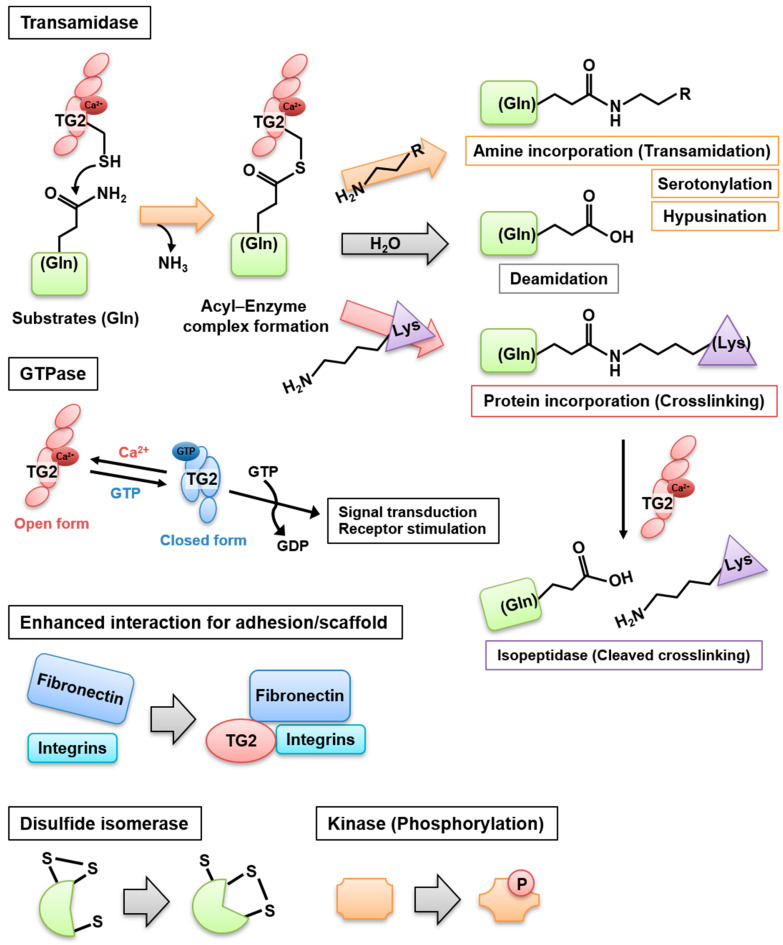
TG2 multifunctional roles. TG2 contributes to the posttranslational modification of several substrate proteins via multiple mechanisms, acting as a transamidase, deamidase, crosslinking protein, isopeptidase, GTPase, adhesion/scaffold protein, disulfide isomerase, and kinase. The mechanisms of some TG2 functions remain unclear. Thanks to its multiple roles, TG2 exerts various biological functions depending on the stimulus, leading to cell death or survival and tissue repair.

**Figure 2 cells-10-01842-f002:**
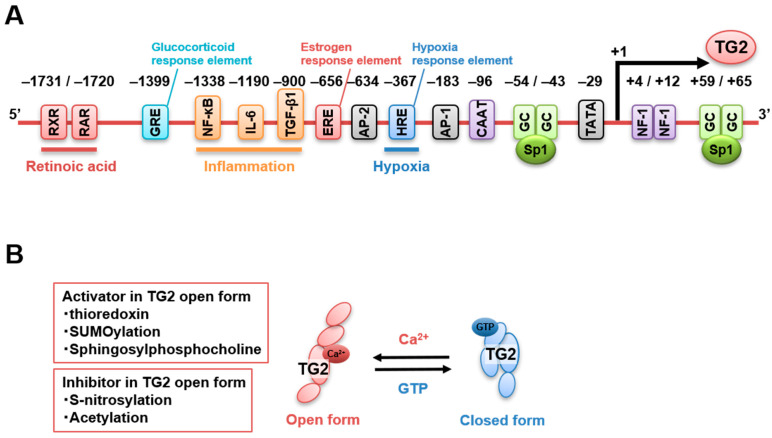
Regulation of TG2 expression and activity. (**A**) Several factors bind to the *TGM2* promoter and regulate *TGM2* expression. Response elements binding sites for retinoic acid, RXR/RAR (−1731 bp and −1720 bp), glucocorticoids, GRE (−1399 bp), NF-κB (−1338 bp), IL-6 (−1190 bp), TGF-β1 (−900 bp), estrogens, ERE (−656 bp), activator protein-2, AP-2 (−634 bp), hypoxia: HRE (−367 bp), activator protein-1, AP-1, and nuclear factor-1 (+4 bp, +12 bp) as well as motif regions such as CAAT box (−96 bp), TATA box (−29 bp), GC box, (Sp1, −54 bp, −43 bp, +59 bp, +65 bp) are indicated. (**B**) The balance between open and closed TG2 structural conformations is mainly regulated by Ca^2+^ and GTP concentrations. In the open conformation, TG2 transamidase activity is enhanced by thioredoxin, SUMOylation, and membrane lipids (sphingosylphosphocholine), whereas it is inhibited by S-nitrosylation and acetylation.

**Figure 3 cells-10-01842-f003:**
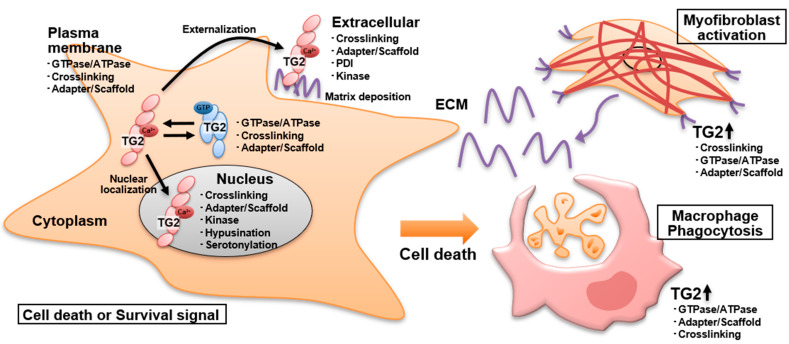
Cellular distribution of TG2 multiple functions and cell types involved in tissue injury processes. Enzymatic activities such as crosslinking, amine incorporation including hypusine and serotonin, GTPase/ATPase, PDI, and kinase as well as non-enzymatic adapter/scaffold activities are shown for TG2 localized in the nucleus, cytosol, underneath the plasma membrane, and in the extracellular compartment. TG2 functions in macrophage engulfment and tissue repair events following cell death induction are also demonstrated.

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
