# Peer review of "Role of Transglutaminase 2 in Cell Death, Survival, and Fibrosis"

_cells, 2021, doi:10.3390/cells10071842_

Round 1

Reviewer 1 Report

This manuscript by Tatsukawa and Hitomi provides a review on the cellular biochemistry of transglutaminase 2, which has been implicated in the development of a wide variety of disease states. The paper is well written and addresses an important topic. I have a few suggestions which may need further attention by the authors:

  1. Line 45, the authors emphasize the focus on immune cell activation. No discussion whatsoever has however been provided on the humoral and T-cell mediated immune responses to TG2.
  2. The clinical implications should be discussed in greater depth. conditions such as celiac disease deserve further attention.
  3. Autoimmunity to TG2 could also precipitate alcoholic liver injury.
  4. The authors may also want to provide a more detailed account on TG2 and endothelial cell pathology.
  5. The existing data on the associations between TG2 and dietary factors might be worth mentioning.
  6. In the Discussion on collagen and crosslinking, the role of lysyl oxidase should be considered.

Minor points

  1. Title: why mention both cell death and survival?
  2. Line 311. ”Profibrotic TGF-β”? Others think TGF-β is antifibrotic, please clarify
  3. page 335, ”fibrotic levels” should be reworded.
  4. References: considering this active area of research, the majority of references cited are rather old. Please update, as appropriate.
  5. Some inconsistencies in reference format: e.g. no 10, all authors capitalized?

Author Response

Point by point response

Reviewer1

Comments and Suggestions for Authors

This manuscript by Tatsukawa and Hitomi provides a review on the cellular biochemistry of transglutaminase 2, which has been implicated in the development of a wide variety of disease states. The paper is well written and addresses an important topic. I have a few suggestions which may need further attention by the authors:

We thank for your comment. As described below, we revised this manuscript according to your comments.

Line 45, the authors emphasize the focus on immune cell activation. No discussion whatsoever has however been provided on the humoral and T-cell mediated immune responses to TG2.

In this review, we summarize and discuss the role of TG2 and relevant studies that we have been conducting so far. Therefore, we did not focus the researches about general immune responses of TG2 although we additionally described a few lines of sentence and references about the other immune cells (Line 292). At previous line 45, we modified “immune cell activation” to “macrophage activation”.

The clinical implications should be discussed in greater depth. conditions such as celiac disease deserve further attention.

Thank you for constructive comments. As you mentioned, we added the description about the clinical implications of TG2 inhibitor for celiac disease (Lines 294 and 403).

Autoimmunity to TG2 could also precipitate alcoholic liver injury.

In accordance with your suggestion, we added the information about relationship between autoimmunity to TG2 and liver injury (Line 296).

The authors may also want to provide a more detailed account on TG2 and endothelial cell pathology.

Thank you for valuable comments. As you mentioned, TG2 has various roles in endothelial cells and blood vessels formation by regulating the stability, migration, spreading, apoptosis, and angiogenesis. The relationship between TG2 and endothelial cell migration was also involved in the fibrosis development. However, in this review, we would like to focus the relevant research that we have been conducted, such as cell death, macrophage activation, and fibrosis.

The existing data on the associations between TG2 and dietary factors might be worth mentioning.

As suggested, we added the description about the food intake of gluten (Line 294).

In the Discussion on collagen and crosslinking, the role of lysyl oxidase should be considered.

Thank you for constructive comments. As you mentioned, we added the description about the role of lysyl oxidase in crosslinking of collagen (Line 325).

Minor points

Title: why mention both cell death and survival?

We included controversial both functions because TG2 has multifunctional role and causes opposing phenotypes such as cell death and survival, which can lead to improvement or worsening of several diseases. However, as the reader might be confused, we decided to remove “/survival”.

Line 311. ”Profibrotic TGF-β”? Others think TGF-β is antifibrotic, please clarify

We believe that TGF-β mediates the fibrosis induction in most cases and it is described as a profibrotic factor in most reviews.

page 335, ”fibrotic levels” should be reworded.

Thank you for comments. We modified “fibrotic” to “fibrosis”.

References: considering this active area of research, the majority of references cited are rather old. Please update, as appropriate.

Thank you for constructive comments. Although we remained the references of original research papers, we modified or added new references as appropriate.

Some inconsistencies in reference format: e.g. no 10, all authors capitalized?

As mentioned, we modified them.

Reviewer 2 Report

This manuscript was well written, and gave a deeply impressions to the reader.

There are some comments below:

  1. Giving an abbreviation table for this manuscript.
  2. Could authors describe the application of TG2 to the commercial?
  3. How about the stability of TG2 under this treatment condition?

Author Response

Point by point response

Reviewer2

Comments and Suggestions for Authors

This manuscript was well written, and gave a deeply impressions to the reader.

We thank for your comment. As described below, we revised this manuscript according to your comments.

There are some comments below:

Giving an abbreviation table for this manuscript.

We added the abbreviation (Line 421).

Could authors describe the application of TG2 to the commercial?

Thank you for comments. TG2 is used commercially to improve the texture of foods and the durability of clothing fibers. It can also be used to make modifications of chemical compounds-linked Gln residues to primary amines for specific labelling of proteins and DNA. In the clinical field, inhibitors of TG2 are being developed for the treatment of coeliac disease and are currently in phase II clinical trials. Such TG2 inhibitors are also currently being developed for the treatment of fibrosis. As suggested, we added this description in the Conclusions and prospects (Line 403).

How about the stability of TG2 under this treatment condition?

Carbone et al. reported that the half-life of TG2 in colorectal cancer HT29 cells is about 10 h. As suggested, we added this information about the stability of TG2 (Line 142).